# Predicting Soil Respiration from Plant Productivity (NDVI) in a Sub-Arctic Tundra Ecosystem

**Olivia Azevedo** [1,*], **Thomas C. Parker** [1], **Matthias B. Siewert** [2] and **Jens-Arne Subke** [1]

1 Biological and Environmental Sciences, University of Stirling, Stirling FK9 4LA, UK; t.c.parker@stir.ac.uk (T.C.P.); jens-arne.subke@stir.ac.uk (J.-A.S.)

2 Department of Ecology and Environmental Science, Umeå University, SE-901 87 Umeå, Sweden; matthias.siewert@umu.se

* Correspondence: olivia.azevedo@stir.ac.uk

**Abstract:** Soils represent the largest store of carbon in the biosphere with soils at high latitudes containing twice as much carbon (C) than the atmosphere. High latitude tundra vegetation communities show increases in the relative abundance and cover of deciduous shrubs which may influence net ecosystem exchange of $CO_2$ from this C-rich ecosystem. Monitoring soil respiration ($Rs$) as a crucial component of the ecosystem carbon balance at regional scales is difficult given the remoteness of these ecosystems and the intensiveness of measurements that is required. Here we use direct measurements of $Rs$ from contrasting tundra plant communities combined with direct measurements of aboveground plant productivity via Normalised Difference Vegetation Index (NDVI) to predict soil respiration across four key vegetation communities in a tundra ecosystem. Soil respiration exhibited a nonlinear relationship with NDVI ($y = 0.202e^{3.508\,x}$, $p < 0.001$). Our results further suggest that NDVI and soil temperature can help predict $Rs$ if vegetation type is taken into consideration. We observed, however, that NDVI is not a relevant explanatory variable in the estimation of SOC in a single-study analysis.

**Keywords:** Abisko; $CO_2$ flux; LAI; modelling; plant functional type; SOC; vegetation index

## 1. Introduction

The Arctic is now the New Arctic [1–4], emerged from a rapidly changing ecosystem as global temperatures continue to rise [5–7]. Temperatures in the region have climbed 0.6 °C per decade over the last 30 years alone [8], growing three times faster than the global mean surface temperature [9]. The Arctic's vulnerability to climate change is of special interest because it contains around half of the total global soil organic carbon (SOC) with most of its ~1300 Pg of SOC stored in permafrost [10–12]. Warming temperatures, expansion of shrub communities and widespread thaw of permafrost have emphasised the urgent need to understand soil carbon dynamics in the region [13,14].

A critical obstacle when determining regional carbon budget lies in the use of detailed data from a limited number of sites to predict the activity of the larger landscape [15,16]. Satellite remote sensing technologies have made it possible to determine vegetation productivity of large and inaccessible areas [17]. The multi-spectral remote sensing from Earth-orbiting satellites is indicating a spectral 'greening' trend of the aboveground component of tundra vegetation [18] due, in part, to changes in community composition driven by shrub expansion and changes in plant traits such as height, leaf area or phenology [19]. Documentation of the 'Arctic greening' phenomenon relies strongly on the use of remotely sensed proxies of vegetation [18,20,21], in particular Normalized Difference Vegetation Index (NDVI), which measures the relative density and health of vegetation for each pixel in a satellite image. There is a known association between NDVI and gross ecosystem productivity [22]. Yet, NDVI values are highly susceptible to a variety of ground-cover fluctuations that can be hard to unravel, including vegetation biomass and type, litter cover

and non-vegetation changes such as unfavourable atmospheric conditions [19]. Further, scaling effects can lead to underestimation of ecosystem productivity parameters due to their non-linear relationship with NDVI and the heterogeneity of Arctic land covers [23]. The high fragmentation and high diversity of Arctic land covers and plant community composition must be taken into account in vegetation monitoring [23–26].

A key part in determining the role of the Arctic in the global climate system is by understanding the dynamism of vegetation [27,28]. Carbon stocks in arctic soils commonly far exceed carbon stored in vegetation [29], and changes in plant ecology impact on physical, chemical and biological processes and feedbacks within the carbon and hydrological cycles [30]. Nevertheless, there is still uncertainty regarding the effect of vegetation change at high latitudes, because seemingly contradictory studies [31] show that it may partly offset atmospheric $CO_2$ increases via increased vegetation growth or it could aggravate soil carbon loss due to higher decomposition rates [32]. Conversely, the focus of research has been mostly on the visible effect of a warming climate or how this reflects on aboveground productivity [16,33,34]. The impact on belowground soil variability and plant productivity in tundra environments remains less clear [34,35]. Iversen et al. (2015) [36] adeptly named plant roots in Arctic tundra "the unseen iceberg", inspired by the emphasis on the importance of leaf and canopy properties, whilst largely ignoring belowground productivity's role.

The structure and function of plant communities are crucial drivers of carbon exchange dynamics [37], with soil $CO_2$ efflux (also soil respiration or R$s$) patterns reflecting plant species differences in litter quality, root production and root respiration [38,39]. Thus, it is imperative to solve how best to measure change in various vegetation types to depict those exchanges in the wider context of climate change impacts on northern landscapes. Plant functional types (PFTs) have been adopted by global ecological modellers to represent broad groupings of functionally similar plant types sharing analogous characteristics and roles [40,41]. Although recent observations indicate the widespread "greening" of the Arctic tundra at a landscape scale, distinct responses are observed among PFTs due to their idiosyncratic physical, biological, and chemical characteristics [36,41]. PFTs' response to warming diverge in terms of productivity, biomass allocation and root distribution as well as their plasticity [34] whose impact on soil carbon will also differ.

Models aiming to project future climate scenarios must capture the uniqueness of the tundra plant community structure and function from observations on the ground. Estimating, or at least constraining flux magnitudes of soil and ecosystem respiration based on remotely sensed vegetation characteristics remains a challenge, as we so far lack a firm understanding of the relationship between aboveground vegetation characteristics and belowground processes.

Our study aims to establish an association between aboveground plant productivity and belowground soil respiration to derive crucial information for tundra carbon budgets. Specifically, we test two hypotheses:

**Hypothesis 1.** *In tundra ecosystems, soil respiration is more strongly related to differences in community-dominant plant functional type than variations in NDVI in each community.*

**Hypothesis 2.** *Organic horizon soil carbon storage is strongly positively related to plant communities.*

## 2. Materials and Methods

### 2.1. Site Description and Vegetation Survey

The research site is situated about 5 km south of Abisko in northern Sweden, around 200 km north of the Arctic Circle (68°19′N, 18°49′E) and c. 500–550 m above sea level. According to the Abisko Scientific Research Station [42], which has been recording temperatures for around 100 years, the average annual temperature for the area is approximately 0 °C with a mean temperature in July of 11 °C (Abisko Science Station). In the last century, the average monthly precipitation in July is 59 mm. We opted for a focused measuring cam-

paign during the peak growing season [23,43] for this first investigation into the correlations between NDVI and belowground biochemical activities. This study took place between the 19th and the 31st of July 2018, which registered monthly air temperatures above the average for this time of the year (14.9 ± 3.9 °C) and lower than typical precipitation levels (42.5 mm).

Our study area is located at the transition between mountain birch forest and open tundra, forming an island treeline ecotone. The vegetation in the area is varied and ranging from snow bed communities, bryophytes and lichens, heath to tall shrubs and birch forest ecosystems [44]. The area was surveyed for four specific tundra plant communities that are important at the regional scale and are representative for large areas across the pan-Arctic region [45]: (1) dense shrubs of *Betula nana* L. (dwarf birch) and (2) *Salix* spp. species (willow), (3) tundra heath dominated by *Empetrum nigrum* L. (crowberry) and crusted areas of (4) lichen heath. Within these, three levels of relative "greenness" per plant community were visually assessed and selected based on the % of plant cover in a gradient from low (<50%), medium (50–70%) to high greenness (75–100%) within 1 m$^2$ plots. Four blocks of ~100 m$^2$ each were created in a tundra area of approximately 3 km$^2$ (Figure 1) acting as spatial replicates within which 12 plots of four vegetation types and three levels of greenness were located. This resulted in a total of 48 measuring locations.

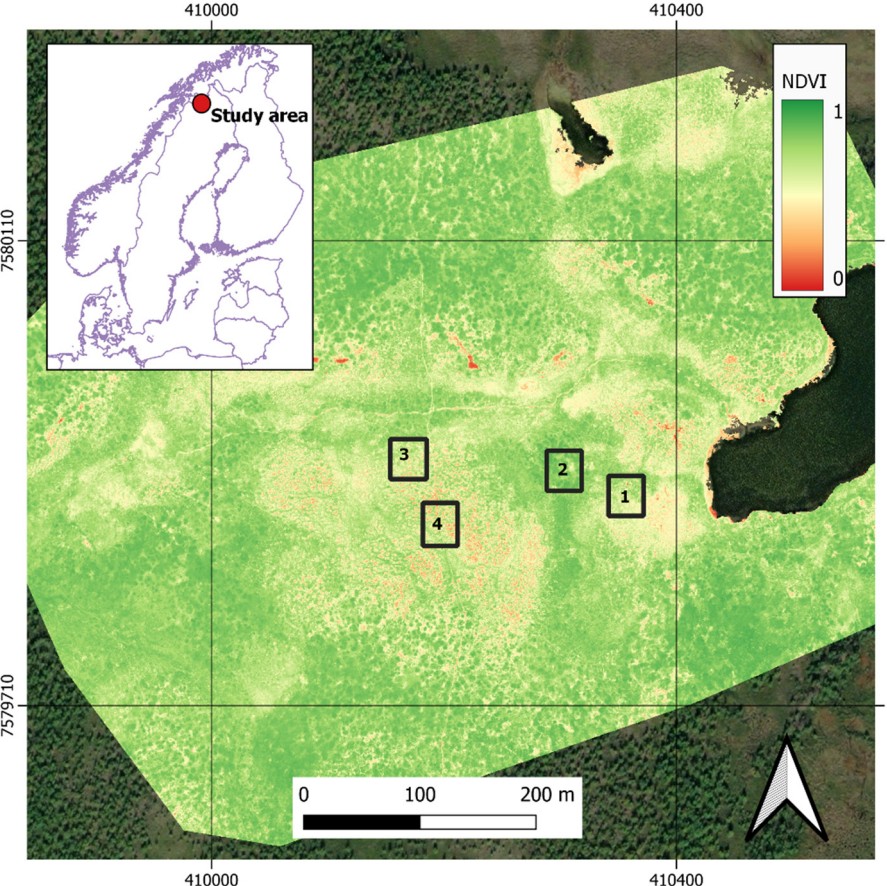

**Figure 1.** Map of the study area. NDVI measurements captured from an UAV (Unmanned Aerial Vehicle) equipped with multispectral sensor (see Siewert and Olofsson 2020) with the superimposed areas of the four sampling blocks (black squares). Green denotes high NDVI while red represents low NDVI or levels of greenness. All plant functional types occur in each block.

## 2.2. NDVI and Land Surface Greenness

NDVI measurements of each 1 m$^2$ plot were carried out with a handheld Spectrosense 2+ spectrometer (Skye Instruments Ltd., Llandrindod Wells, UK) with 2 channels

(650/40 nm and 800/40 nm centre wavelengths/bandwidth). The NDVI vegetation index is calculated according to:

$$NDVI = (NIR - Red)/(NIR + Red)$$

where NIR is reflectance in the near-infrared (800 nm centre wavelength) and Red is reflectance in the red range of wavelengths (560 nm) of the light spectrum, resulting on a scale between $-1$ and 1, where 0 is typically bare ground with minimal to no vegetation greenness and 1 is high degree of greenness (Figure 1). NDVI has been shown to be a good proxy for LAI (leaf area index), and Van Wijk and Williams (2005) [46] successfully established a relationship between NDVI and LAI in a tundra environment ($R^2 = 0.73$), using comparable sensors and wavelengths to ours (580 to 690 nm and 725 to 1100 nm, respectively). We apply their equation to convert our own NDVI values into LAI accordingly: $LAI = 0.003 \, e^{7.845 \times NDVI}$.

### 2.3. Soil Respiration

To measure soil respiration, PVC (Polyvinyl chloride) collars with a diameter of 15 cm and a height of 7 cm were placed on the soil surface taking care to gently clear the diameter space of any vegetation or plant debris. These were sealed to the soil using a non-setting putty. Soil $CO_2$ efflux measurements were taken using a portable Infra-Red Gas Analyser (IRGA, EGM-4, PP Systems International, Amesbury, MA, USA) equipped with CPY-4 chamber, which was carefully placed on top of the PVC collar, ensuring a good seal between chamber and collar.

$CO_2$ captured by the chamber is the total soil $CO_2$ efflux, including microbial and plant root activity, with respiration rates being calculated based on a linear function of the increase in the amounts of carbon dioxide within the compartment over a period of 90 s [47]. Entire blocks were measured on the same day, avoiding significant differences in temperature and moisture over time. Measurements from all collars took place on three occasions during the 10-day period, and the order in which the four blocks and the 12 plots within each block were sampled was randomised. To explain variations in soil respiration, soil temperature at 5 cm depth was also recorded between 10 and 20 cm from each soil flux collar every time $CO_2$ efflux measurements took place.

### 2.4. Soil Cores: Moisture, Organic Matter, and Elemental Analysis

Three soil cores per plot were collected by inserting a 4.5 diameter × 20 cm deep soil corer into the ground within each plot until it would not go any deeper. The assumption is that the corer hit the parent material or a large rock. Organic soil horizon was separated from mineral soil and samples from all three cores per plot were combined according to horizon type. Soil samples were utilised to determine soil organic matter and soil moisture, because soil moisture (like soil temperature) can influence variation in R$s$ rates. Soil samples were sieved with a 2 mm sieve to remove the coarse fraction and then weighed before and after being dried at 60 °C until all moisture had been removed. This was verified by regularly reweighing the sample to see if the weigh was still being lost. Fine roots (<2 mm) were picked from the soil samples and their weight was recorded to estimate root biomass (in g of roots per m$^2$).

Soil organic matter content was quantified by loss-on-ignition (LOI; [48]) by placing oven dried soil samples in a muffle furnace at 450° C for a minimum of 12 h. The organic matter amount is estimated as the mass difference of samples before and after incineration in the furnace. To calculate SOC content, approximately 5 g of soil selected from 23 samples (14 organic and 9 mineral soil) under high level of greenness were also milled to powder. The C content of sub-sample of c. 3 mg was determined by elemental analysis (Flash*Smart* 2000 NC Org., Thermo Fisher Scientific, Cambridge, UK).

### 2.5. Statistical Analysis

All analyses were carried out on R version 4.0.2 (22 June 2020). Log transformation was employed where data were not normally distributed upon analysis of model's distribution of the residuals. Homogeneity of variance was tested by applying the ANOVA() function (a variation of Levene's Test). The aov() function was employed to determine if there were statistically significant differences between the respiration and SOC means of the respective plant communities. A post-hoc Tukey test was carried out using the TukeyHSD() ('Honest Significant Difference') function to perform a multiple pairwise-comparison between the means of groups. We report F values (ratio of variance between and within groups) and level of significance ($p > 0.05$, $p < 0.05$, $p < 0.01$, or $p < 0.001$).

To address hypothesis 1, a linear mixed-effects model was fitted first with maximum likelihood (ML) using the lme4 package in R to understand the effect of NDVI, soil moisture, soil temperature, fine roots as fixed effects on soil respiration [49,50]. "Block" and "Community" (representing spatial replicates and plant functional types, respectively) within the experiment were identified as random factors. As LAI is derived mathematically from measured NDVI values, it is not an independent parameter and not considered in the model. The soil $CO_2$ efflux response to soil temperature (at 5 cm depth) was derived from model parameters and expressed as $Q_{10}$, i.e., the factor by which $CO_2$ flux increases following a temperature increase by 10 °C.

We tested the relationship between carbon in the organic horizon (SOC) against NDVI, soil moisture, fine roots, and soil temperature to address hypothesis 2, by using a linear mixed-effects model with Block and Community once again explaining random variance. Following model simplification by AIC (Akaike Information Criterion) to estimate model parsimony (where the model with the lowest AIC is the most parsimonious), the final version was fitted with restricted maximum likelihood (REML) for more accurate coefficient estimates. We also calculated the ICC (intraclass correlation) which can be interpreted as the proportion of the variance in Block and plant community (our random effects) that can be explained by the grouping structure in the population. The ICC is calculated by dividing the random effect variance, $\sigma^2_i$, by the total variance, i.e., the sum of the random effect variance and the residual variance, $\sigma^2_\varepsilon$ [51].

## 3. Results

### 3.1. Organic Matter and Carbon

Organic matter content results from loss-on-ignition correlated significantly with C content established by the elemental analysis across both mineral and organic soil horizons ($y = 0.4472x + 0.7528$, $R^2 = 0.99$, $p < 0.001$). This relationship was applied to organic matter results to obtain C content for all plots. At $5.63 \pm 4.51$ kg m$^{-2}$, stocks of carbon are greater in the organic horizon of the soil profile across all vegetation types than in the mineral horizon ($1.28 \pm 0.82$ kg m$^{-2}$). Focusing only on the organic horizon (Table 1), willow has the highest level of SOC ($7.19 \pm 6.33$ kg m$^{-2}$) while lichen has the lowest concentrations ($3.09 \pm 1.83$ kg m$^{-2}$). Betula nana presented the least extent of relative data variation (Table 1).

**Table 1.** Descriptive statistics for plant communities, soil respiration (µmol m$^{-2}$ s$^{-1}$), vegetation index, soil organic carbon (SOC), organic matter stocks in the organic horizon (kg m$^{-2}$), fine roots (g m$^{-2}$) and abiotic conditions (soil moisture %, soil temperature °C). Some cores under *E. nigrum* vegetation did not show an organic horizon.

|  | *B. nana* (N= 12) | *E. nigrum* (N= 10) | Lichen (N= 12) | Willow (N= 12) | Overall (N= 46) |
|---|---|---|---|---|---|
| **Fine Roots** |  |  |  |  |  |
| Mean (SD) | 146 (87.8) | 146 (87.8) | 146 (87.8) | 146 (87.8) | 93.2 (77.4) |
| Median [Min, Max] | 125 [42.8, 216] | 125 [42.8, 216] | 125 [42.8, 216] | 125 [42.8, 216] | 78.9 [14.0, 316] |
| **LAI** |  |  |  |  |  |
| Mean (SD) | 1.82 (0.563) | 1.13 (0.413) | 0.146 (0.111) | 1.87 (0.276) | 1.25 (0.807) |
| Median [Min, Max] | 1.88 [0.980, 2.72] | 1.11 [0.630, 1.90] | 0.105 [0.040, 0.340] | 1.95 [1.51, 2.31] | 1.41 [0.040, 2.72] |
| **NDVI** |  |  |  |  |  |
| Mean (SD) | 0.813 (0.0427) | 0.751 (0.0431) | 0.461 (0.0961) | 0.818 (0.0221) | 0.709 (0.161) |
| Median [Min, Max] | 0.820 [0.740, 0.870] | 0.750 [0.680, 0.820] | 0.450 [0.340, 0.600] | 0.815 [0.780, 0.850] | 0.780 [0.340, 0.870] |
| **SOM** |  |  |  |  |  |
| Mean (SD) | 13.0 (6.49) | 14.4 (11.4) | 6.70 (3.99) | 15.7 (13.9) | 12.4 (9.98) |
| Median [Min, Max] | 12.0 [5.89, 25.4] | 11.3 [1.64, 39.1] | 5.68 [1.86, 16.2] | 10.9 [5.51, 55.1] | 9.73 [1.64, 55.1] |
| **SOC** |  |  |  |  |  |
| Mean (SD) | 5.71 (2.47) | 6.72 (5.29) | 3.09 (1.83) | 7.19 (6.33) | 5.63 (4.51) |
| Median [Min, Max] | 5.42 [2.69, 11.6] | 5.17 [0.780, 17.8] | 2.62 [0.990, 7.43] | 4.97 [2.54, 25.1] | 4.66 [0.780, 25.1] |
| **Soil Moisture** |  |  |  |  |  |
| Mean (SD) | 63.0 (7.32) | 57.3 (13.0) | 44.0 (14.3) | 62.9 (12.0) | 56.8 (14.0) |
| Median [Min, Max] | 64.4 [49.4, 72.1] | 61.5 [28.3, 67.8] | 43.7 [22.8, 63.2] | 65.3 [31.7, 75.0] | 60.8 [22.8, 75.0] |
| **Soil Respiration** |  |  |  |  |  |
| Mean (SD) | 5.30 (2.31) | 2.98 (1.05) | 1.26 (0.340) | 5.87 (2.27) | 3.89 (2.54) |
| Median [Min, Max] | 5.41 [1.68, 9.93] | 2.71 [1.26, 4.41] | 1.29 [0.770, 1.86] | 5.75 [2.68, 10.2] | 3.19 [0.770, 10.2] |
| **Soil Temperature** |  |  |  |  |  |
| Mean (SD) | 10.9 (1.22) | 12.9 (1.31) | 16.6 (1.38) | 10.7 (0.883) | 12.8 (2.69) |
| Median [Min, Max] | 11.0 [8.43, 12.8] | 13.3 [11.1, 14.9] | 16.2 [14.6, 18.7] | 10.9 [8.93, 11.8] | 11.8 [8.43, 18.7] |

*3.2. Soil Respiration, Organic Carbon, and Vegetation Indices*

Different plant communities are not associated with significantly different stocks of SOC and show high variability (Figure 2A; F = 2.08, *p* > 0.05). However, plant communities are found to have significantly different rates of soil respiration (Figure 2B; F = 18.04, *p* < 0.001), caused mainly by reduced soil respiration under Empetrum nigrum and lichen vegetation compared to more productive shrub type vegetation (post-hoc Tukey test; Figure 2B), and show less variation compared to organic carbon.

Soil respiration exhibited a nonlinear relationship with NDVI (Figure 3A; [y = 0.202 e$^{3.508\,x}$, *p* < 0.001]). The regression indicates a doubling in Rs for an increase in NDVI of 0.1976. The relationship between soil C in the organic horizon and NDVI is only marginally significant (Figure 3C [y = 0.035 + 7.897x, R$^2$ = 0.08, *p* = 0.057]). Conversion of NDVI to LAI (y = 0.003e$^{7.845x}$, Van Wijk and Williams, 2005), resulted in a significant linear relationship with soil respiration (Figure 3B [y = 0.982 + 2.332x, R$^2$ = 0.55, *p* < 0.001]) but not for soil organic carbon (Figure 3D [y = 4.023 + 1.291x, R$^2$ = 0.05, *p* > 0.05]).

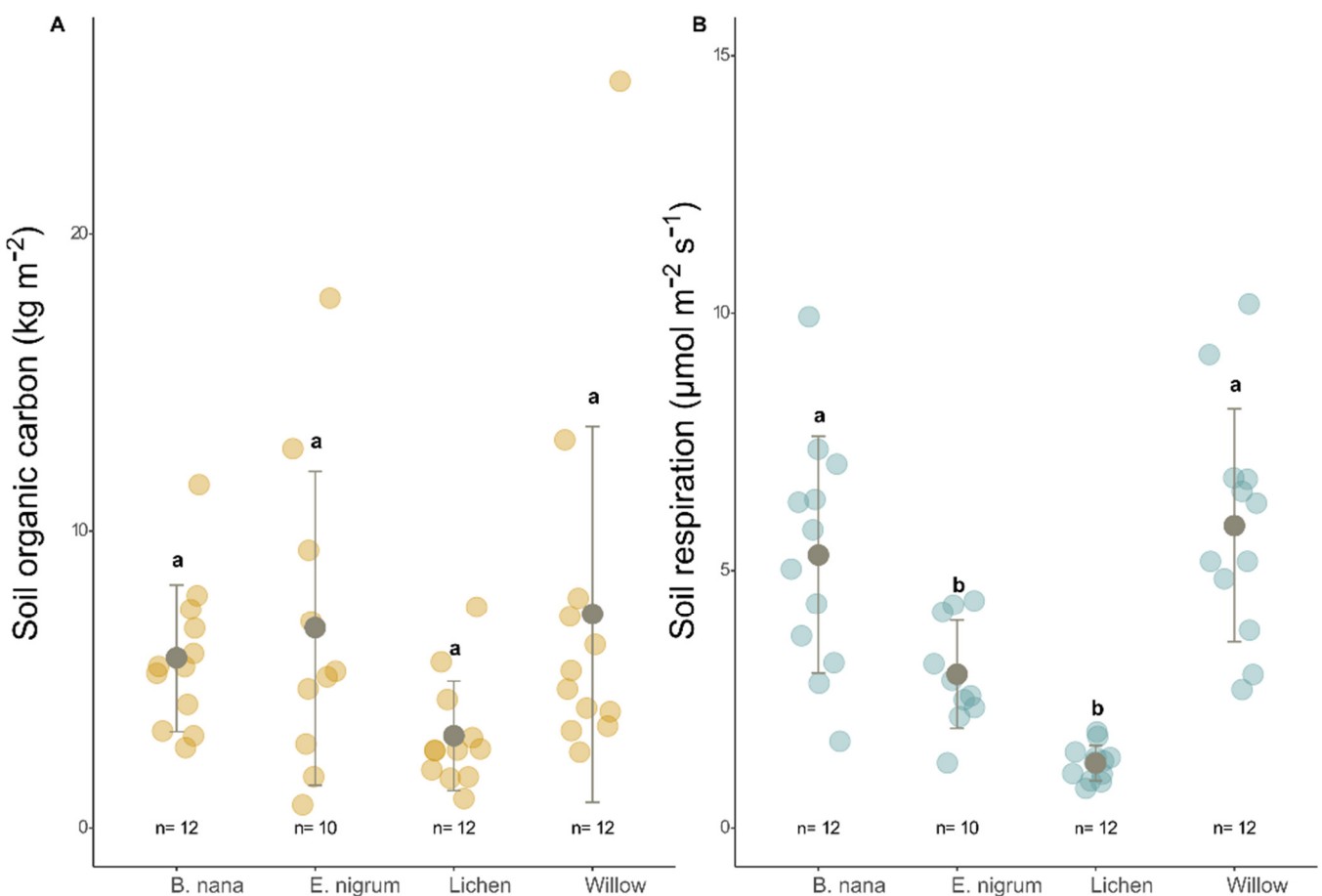

**Figure 2.** Distribution of soil organic carbon (**A**) and soil respiration (**B**) among plant communities. Summary statistics (mean and standard deviation; grey colour) are superimposed onto raw data. Dissimilar lower-case letters within each panel indicate significant differences within each data set according to a post-hoc Tukey test.

The parameter estimates (Table 2) for the mixed effects model indicate that for an increase in soil temperature by 1 °C, there is a 6.6% increase in Rs ($p < 0.001$; $Q_{10} = 1.90$) and that for a 0.1 rise in NDVI, we observe a 24.6% increase in soil respiration ($p < 0.01$).

**Table 2.** Parameter estimates for both final mixed effects models, with standard error, level of significance, 95% confidence intervals (CI) and intraclass correlation (ICC). ICC represents the effect of the block design and plant community on the model.

|  |  | **Parameter Values** | **Standard Error** | **CI (Lower)** | **CI (Upper)** |
|---|---|---|---|---|---|
|  | NDVI | 2.20 ** | 0.81 | 0.58 | 4.32 |
| **log (R*s*)** | Soil temperature | 0.0644 *** | 0.0191 | 0.0247 | 0.1010 |
|  | Constant | −1.38 | 0.73 | −3.04 | 0.055 |
|  | ICC (adjusted) | 0.495 |  |  |  |
|  | Soil moisture | 0.0264 *** | 0.0055 | 0.0154 | 0.0372 |
| **log (SOC)** | Fine roots | 0.00205 * | 0.00101 | 0.00010 | 0.00407 |
|  | Constant | −0.209 | 0.340 | −0.863 | 0.454 |
|  | ICC (adjusted) | 0.414 |  |  |  |

Note: * $p < 0.05$; ** $p < 0.01$; *** $p < 0.001$.

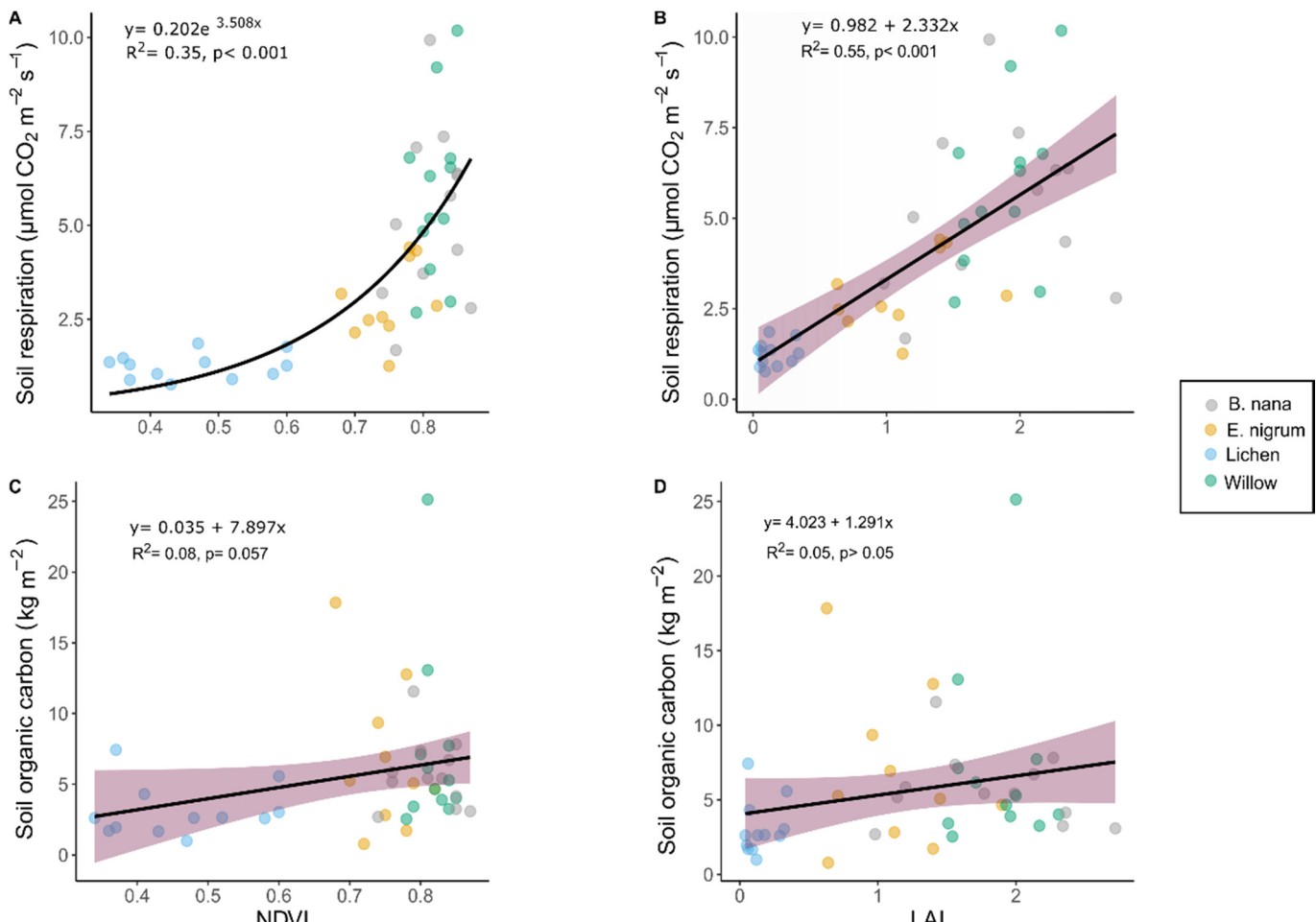

**Figure 3.** Relationship of soil respiration (**A**,**B**) and soil organic carbon (**C**,**D**) with NDVI and LAI. Graphs show a modelled line between variables (black) with 95% confidence interval (shaded pink area).

While soil moisture did not correlate with R$s$ ($p > 0.05$), there is a highly significant correlation between soil moisture on soil organic carbon ($p < 0.001$), with carbon in the organic horizon predicted to increase by 1.063 kg m$^{-2}$ for every 1% increase in moisture (95% CI = 1.036, 1.089 kg m$^{-2}$) after back-transforming the log values. We also observe a 1.0047 kg m$^{-2}$ increase for every g m$^{-2}$ fine root biomass ($p < 0.05$; 95% CI = 1.0045, 1.0142 kg m$^{-2}$). However, the relationship between SOC and NDVI is not significant ($p > 0.05$). Our models also show that the intra-class correlation that measures the random effect variance for block and plant community is high both for the soil respiration model (ICC = 0.495) and the organic carbon model (ICC = 0.414), indicating that a considerable fraction of the total variation in the data lies between groups.

## 4. Discussion

### 4.1. Predicting Soil Respiration in a Highly Heterogeneous Environment

Accurate measurement of soil respiration rates is crucial for the evaluation of carbon cycling in the Arctic [52]. Our results show that vegetation type and level of greenness (NDVI) are significantly and measurably related to soil respiration. However, this is a multifaceted relationship and cannot be adequately forecast by levels of NDVI alone (Hypothesis 1). Nevertheless, the inclusion of NDVI and soil temperature measurements along with information about plant community composition and spatial distribution resulted in a significant predictive model (Table 2).

Our results echo the intricacy of biogeochemical characteristics in high latitude regions [53,54], reflecting the complexity of an ecotone between boreal forest and tundra.

Arctic tundra soils are typically organic-rich due to the accumulation of plant material and slow decomposition rates [55]. Rising temperatures cause an increase in decomposition by increasing soil microbial activity, impacting particularly on an ecosystem characterised by low primary productivity, low nutrient inputs and slow cycling of these nutrients [56,57]. Soil respiration is highly responsive to temperature increases [58–60], and soil moisture responses have also been widely reported [61,62]. Understandably, abiotic variables such as soil temperature and soil moisture remain a classic approach in regression analysis to determine R$s$, with many empirical relationships having been established via field measurements [63–65]. We found no direct influence of moisture on short-term variation in soil $CO_2$ efflux rates, and we speculate that characteristic variations in soil water content were confounded with vegetation types. With respect to the explanatory power of temperature on the rate of $CO_2$ efflux this is limited in highly heterogenic landscapes with varied vegetation cover [66]. Climate warming favours growth of deciduous shrubs at the cost of other plant types in the arctic tundra such as grasses, mosses and lichens [52,67,68]. Nevertheless, terrestrial vegetation is one of the largest sources of uncertainty in climate change predictions [69,70] and studies investigating the significance of $CO_2$ release from soils in relation to vegetation cover are rare, but necessary to get a full picture of Arctic carbon dynamics [38,43,71]. Most climate models currently use only two PFTs (one grass and one shrub) to embody Arctic vegetation [41,72]. In order to capture the high-latitude vegetation processes, it is important to embrace Arctic and boreal specific PFTs in modelling vegetation change predictions, including deciduous and evergreen shrubs, sedges, grasses, forbs, *Sphagnum* and non-*Sphagnum* mosses and lichens [69,73]. Our data show that different plant communities have different soil respiration rates (Figure 2B), underlining the importance of incorporating plant community composition into the partitioning of energy and carbon fluxes in the Arctic [74].

Recent satellite remote sensing studies, which revealed greening trends in the Arctic associated with increasing temperatures, suggest intensification of productivity and photosynthetic uptake of $CO_2$ [18]. Nonetheless, the study of broad-scale ecosystem level changes in soil respiration as a response to variations in climate and vegetation cover are less straightforward [52]. The evaluation of the global impact of changes in the net carbon balance of high-latitude ecosystems requires the understanding of the controls on belowground $CO_2$ efflux at correct spatial and temporal scales [39]. The implementation of vegetation indices such as NDVI to aid the process of discovery of $CO_2$ efflux patterns is faced with the challenge of the fine-scale variation of the Arctic vegetation, corresponding to differences in topography, hydrology and frost-heaves [15], requiring highly resolved UAV data to obtain a full picture of landscape level distribution of ecosystem productivity variables [23]. A girdling study from a site near to the present study shows that C supply from the canopy drives root and rhizosphere respiration in treeline birch and dense willow shrub patches [43], confirming the significance of C assimilation on belowground rhizosphere processes [75,76]. Our results from tundra vegetation align with these findings, as we also show that higher the productivity of the canopy (assessed by NDVI) correlates with higher soil $CO_2$ flux, most likely associated with rhizosphere C supply. Arctic models already consider conceptual representations of root biomass, depth distribution, turnover and respiration rates, among others. However, few models explicitly explore root morphology, phenology and interactions between roots and their surroundings [36].

Linking R$s$ to NDVI would allow for landscape scale estimates of R$s$ that are so far lacking in the spatial distribution of soil respiration data estimation [77] and research advocates this potential for remote sensing to assess $CO_2$ efflux [77–79]. Our study revealed that a combination of remote sensing products (NDVI) and in situ sampling of soil temperature adequately predicted R$s$. A comparison of several statistical models using remote sensing products such as Land Surface Temperature (LST; the radiative skin temperature of the Earth measured by a sensor like MODIS), root zone soil moisture and photosynthesis Enhanced Vegetation Index (EVI), to examine their relationship with R$s$ [79] to models based on in situ measurements in a deciduous forest ecosystem (Huang, Gu and Niu, 2014)

concluded that models based entirely on spatial data products showed lower explanation capacity for seasonal variation of Rs than the model measured from ground data ($R^2$ = 0.76 and 0.90, respectively). However, we would expect the relationship between Rs and remotely sensed NDVI more straight forward in tundra than in broadleaved forest where the spatial variation is modulated by large trees with extensive root networks.

*4.2. Soil Organic Carbon and Vegetation Indices Constraints*

Associations between soil carbon and vegetation indices are instrumental for understanding the magnitude and acceleration of $CO_2$ emissions in a warming Arctic [38,80,81], because, unlike belowground features, aboveground vegetation can be easily detected remotely using aerial and satellite imagery. SOC storage results from the balance of carbon inputs and losses and is therefore a key regulator of land-atmosphere feedbacks in the face of climate change [82].

We know that more productive plant communities with higher leaf area have higher respiration rates [38], but despite the established positive relationship between leaf and fine root turnover rates with LAI [83] our results show no evidence for NDVI propelling variations in fine root biomass either between or within vegetation types. These results diverge from our findings relating to SOC, where we concluded that soil organic carbon can be adequately explained by levels of soil moisture ($p < 0.001$) and fine roots biomass ($p < 0.05$) in the organic horizon and by the consideration of plant community composition in the model (ICC = 0.414).

Previous research looking at correlations between NDVI and SOC suggested a good level of prediction and modelling accuracy when using time series regression, yet it also observed that single-data NDVI alone cannot be used to forecast SOC [22], but requires more sophisticated model based digital soil mapping approaches in the Arctic, including a range of environmental variables [26,84]. Adding to the multi temporal constraint of NDVI/SOC associations is the element of soil depth evaluation [85]. The functional relationship between SOC and vegetation cover sensitive in certain spectral bands allowed to use spectral remote sensing data in digital soil mapping approaches for robust estimates of SOC [26,86]. However, soils in the tundra show tremendous subsurface variability of SOC caused by annual freeze and thaw processes and permafrost, yet, the lack of surface expression often conceals subsurface variability with valuable information potentially unaccounted for [87]. Moreover, vegetation cover and soil moisture are known to lead to inaccurate predictions of SOC [88,89].

A follow-on study should measure LAI in the field so we can better define the relationships between NDVI/LAI and soil respiration. This would suitably verify the predictive power of LAI measured on the ground on soil respiration. Our study provides strong evidence for the potential to use NDVI as a predictor for soil respiration at local scale and highlights the more complicated relationship between vegetation productivity and SOC.

## 5. Conclusions

Our findings show the usefulness of remotely sensed products to infer biological indicators not only of plant productivity, but also relating it to the dynamics of soil respiration in a highly heterogenic ecosystem. It thus improves our understanding of the interactions between above- and belowground processes in a tundra ecosystem, by successfully linking soil respiration with aboveground plant productivity at a landscape level. Regression analysis determined an empirical relationship between Rs and remotely sensed levels of greenness (NDVI) under distinct vegetation types. The inclusion of a photosynthesis related index such as EVI may possibly solve the classic saturation problem as demonstrated by recent studies in remote sensing. This single-site study was unable to directly link NDVI to SOC for prediction purposes. With the increasing availability of high-resolution multi-spectral satellite and UAV imagery resolving plant communities and productivity at ever finer grain, our results highlight the potential to map and estimate soil respiration at landscape scale, providing crucially needed ground reference data for large scale, but



coarse-resolution satellite validation and model calibration. This may eventually provide urgently needed understanding of soil respiration dynamics in a warming New Arctic.

**Author Contributions:** Conceptualization, T.C.P. and J.-A.S.; methodology, O.A., T.C.P. and M.B.S.; formal analysis, O.A. and J.-A.S.; writing—original draft preparation, O.A.; writing—review and editing, all authors; visualization, O.A., M.B.S.; supervision, J.-A.S., T.C.P.; funding acquisition, J.-A.S. All authors have read and agreed to the published version of the manuscript.

**Funding:** This research was funded by an Altajir Trust scholarship for O.A., the Natural Environment Research Council (NERC) of the UK, grant numbers NE/P002722/1 and NE/P002722/2 to J.-A.S. and T.C.P, and the Carl Tryggers Foundation for Scientific Research, grant number CTS 16: 342 to Johan Olofsson, Umeå University, for M.B.S.

**Institutional Review Board Statement:** Not applicable.

**Informed Consent Statement:** Not applicable.

**Data Availability Statement:** The data presented in this study are openly available at the University of Stirling's online data repository https://datastorre.stir.ac.uk/ (accessed on 29 June 2021).

**Acknowledgments:** We warmly acknowledge technical support for C analyses of soils at the University of Stirling. We thank referees of the original manuscript for constructive comments that have helped to improve this paper.

**Conflicts of Interest:** The authors declare no conflict of interest.

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
