# Peer review of "Predicting Soil Respiration from Plant Productivity (NDVI) in a Sub-Arctic Tundra Ecosystem"

_remotesensing, doi:10.3390/rs13132571_

Round 1

Reviewer 1 Report

Summary:

This paper addresses a crucial topic regarding measuring soil respiration in the Arctic region. The warming temperature and increasing burning have been reported recently. A widespread greening trend has also been observed, while responses and variations of the local carbon cycle are less known. This study combined on-site observations and high-resolution satellite observations to assess soil carbon storage and respiration. The results also approved that the NDVI can be used as an effective proxy to reflect vegetation carbon productivity and measure soil respiration in the Arctic.

While the subject of the paper has merit and important indication, there are a few concerns regarding the methods and results. The description regarding the hypothesis, datasets, and results can be improved. In addition, a few minor expressions can be addressed to improve the readability of the manuscript. Details of major and minor concerns are given below.

Major comments:

  1. When the authors listed the hypothesis, ‘plant community (composition)’ or ‘plant community’ were used. This term was relatively general and ambiguous. Does it refer to land cover type or vegetation type? Or it can also refer to the composition of different plants and the numbers of vegetation types within a small study plot.
  2. In the Method section, the authors set ‘three levels’ of greenness for each plant community (or vegetation type). What are the detail levels? I suggest for each plant type, the levels are different given that they have quite different NDVI values. In addition, it would be appreciated if detailed temporal and spatial resolutions of the NDVI product are available. NDVI is a good measurement and indication of LAI and productivity, is it possible to measure vegetation productivity using available high-resolution LAI products? If so, the results can be better bolstered and compared.
  3. When listing factors that are related to SOC, fine roots are included and mainly measured by weight? Is this a conventional method? For the measurements of root systems are complicated and the length and density of roots can respectively affect the plants a lot and differently.
  4. It is plausible to apply the statistical regression fitting model to estimate different factors’ impacts on SOC and Rs. On the other hand, the results also briefly suggested that there might be interactions among these factors. For example, soil temperature would affect plant physiology and NDVI. Would these interactions amplify or diminish their effects on Rs or SOC? In addition, it is interesting to see that the NDVI is well correlated with Rs but not SOC, then what’s the relationship or association between Rs and SOC? And how to understand the impact of variations in Rs or SOC on the carbon cycle?
  5. The study was mainly focused on July. Is there any certain reason for choosing this month? How the results might be different in a warmer month or colder seasons?
  6. Soil temperature and soil moisture were selected in the fitting model. When considering soil carbon dynamics, other soil properties like porosity and soil composition can also affect biochemistry. Would these factors affect carbon storage in the Arctic region? In addition, the air temperature has been suggested to contribute to the greening in the study region and more related to plant phenology, would including air temperature in the model to improve the estimate of soil respiration?

Minor comments:

Abstract: Line 11, does it refer to ‘observations of widespread increase of …’instead ‘widespread observations of increase…’.

Line 30: ‘have emphasised’

Line 47: ‘due to’

Line 55: Replace ‘surrounding’ with ‘regarding’

Line 63: Does it refer to ‘the importance of leaf and canopy properties to indicate belowground productivity’s role.’

Line 68: Replace ‘recognise’ with ‘figure out’, and replace ‘change’ with ‘changes’.

Line 133: What is the full term of ‘PVC’.

Line 201: Does ‘the mineral horizon’ refer to the vertical distribution of carbon.

Table 1: Does ‘Relationship’ refer to mean values?

Author Response

  1. When the authors listed the hypothesis, ‘plant community (composition)’ or ‘plant community’ were used. This term was relatively general and ambiguous. Does it refer to land cover type or vegetation type? Or it can also refer to the composition of different plants and the numbers of vegetation types within a small study plot.

We now clarify the hypothesis to relate to the dominant plant species, as was our original intention. The hypothesis now reads (Line 90-92): “In tundra ecosystems, soil respiration is more strongly related to differences in community-dominant plant functional type than variations in NDVI in each community (hypothesis 1)”.

  1. In the Method section, the authors set ‘three levels’ of greenness for each plant community (or vegetation type). What are the detail levels? I suggest for each plant type, the levels are different given that they have quite different NDVI values. In addition, it would be appreciated if detailed temporal and spatial resolutions of the NDVI product are available. NDVI is a good measurement and indication of LAI and productivity, is it possible to measure vegetation productivity using available high-resolution LAI products? If so, the results can be better bolstered and compared.

We agree that this text was ambiguous. The ‘three levels of greenness’ were based on a visual assessment of canopy density for each plant functional type. We now write that (Line 114): “…three levels of relative “greenness” per plant community were visually assessed and selected based on the % of plant cover in a gradient from low (<50%), medium (50-70%) to high greenness (75-100%) within 1 m2 plots.”

  1. When listing factors that are related to SOC, fine roots are included and mainly measured by weight? Is this a conventional method? For the measurements of root systems are complicated and the length and density of roots can respectively affect the plants a lot and differently.

We agree that there are multiple methods for measuring a variety of root traits. Root biomass is a basic but important measurement variable (See Hartley et al. 2012, https://doi.org/10.1038/nclimate1575; Sloan et al. 2013, https://doi.org/10.1111/gcb.12322.). It was not in the scope of the paper to go further into root traits, although that would be valuable for further research. To avoid any misunderstanding, we further explained in the methods section that the weight of the roots was used to calculate root biomass (Line 165-167): “Fine roots (< 2mm) were picked from the soil samples and their weight was recorded to estimate root biomass (in g of roots per m2).”

  1. It is plausible to apply the statistical regression fitting model to estimate different factors’ impacts on SOC and Rs. On the other hand, the results also briefly suggested that there might be interactions among these factors. For example, soil temperature would affect plant physiology and NDVI. Would these interactions amplify or diminish their effects on Rs or SOC? In addition, it is interesting to see that the NDVI is well correlated with Rs but not SOC, then what’s the relationship or association between Rs and SOC? And how to understand the impact of variations in Rs or SOC on the carbon cycle?

The referee raises an interesting point linking soil temperature to plant physiology, SOC and respiration. In our analysis, we included soil temperature in both models, but it was eliminated from the final version for the SOC model via model simplification, as it is standard statistical practice (see Crawley 2007, DOI:10.1002/9780470515075). Given the low degree of significance for correlations linking soil temperature to SOC, NDVI and physiology, we avoided speculation on these based on our data. To establish possible links between these (and ultimately plant/soil C cycling) would require a more detailed investigation of respiration response, possibly involving temperature manipulations, which were not part of our investigation.

  1. The study was mainly focused on July. Is there any certain reason for choosing this month? How the results might be different in a warmer month or colder seasons?

We chose the peak growing season for this first investigation into the connections between NDVI and belowground processes. We did aim to provide a complete CO2 flux and NDVI overview of the growing season, although this would be a useful extension for this work. We hope that clarifying the measuring period in ‘Methods’ helps to explain the measuring approach which is consistent with the objectives of our investigation.  (Line 101-103): “We opted for a focused measuring campaign during the peak growing season [23, 43] for this first investigation into the correlations between NDVI and belowground biochemical activities. This study took place between the 19th and the 31st of July 2018…”.

  1. Soil temperature and soil moisture were selected in the fitting model. When considering soil carbon dynamics, other soil properties like porosity and soil composition can also affect biochemistry. Would these factors affect carbon storage in the Arctic region? In addition, the air temperature has been suggested to contribute to the greening in the study region and more related to plant phenology, would including air temperature in the model to improve the estimate of soil respiration?

Soils are predominantly organic-rich, mostly on ericaceous peat with low bulk density. Underlying mineral soil is fine-textured (rich in silt), and it is possible that different textures near the surface would influence C storage and flux. The sub-arctic tundra is dominated by organic soils, and the different dominant vegetation types we focus on show no systematic bias in soil conditions that would affect our conclusions.

Minor Comments:

Abstract: Line 11, does it refer to ‘observations of widespread increase of …’instead ‘widespread observations of increase…’. We rewrote the abstract in response to the observations from other referees, and the expression in line 11 was completely removed.

Line 30: ‘have emphasised’ Replaced “has emphasised” with “have emphasised”.

Line 55: Replace ‘surrounding’ with ‘regarding’. We have replaced “surrounding” with “regarding”.

Line 63: Does it refer to ‘the importance of leaf and canopy properties to indicate belowground productivity’s role.’ Thank you for pointing out the unclear wording. We have changed this to illustrate the fact that remote sensing predominantly focuses on aboveground productivity, when belowground productivity and C cycling are at least as important. The sentence now reads (line 67/68) "...the importance of leaf and canopy properties, whilst largely ignoring belowground productivity’s role."

Line 68: Replace ‘recognise’ with ‘figure out’, and replace ‘change’ with ‘changes’. Replaced “recognise” with “solve”; corrected “changes”.

Line 133: What is the full term of ‘PVC’. Added “(Polyvinyl chloride)”.

Line 201: Does ‘the mineral horizon’ refer to the vertical distribution of carbon. It refers to the vertical distribution of carbon.

Table 1: Does ‘Relationship’ refer to mean values? It now reads: “Descriptive statistics for…”

Finally, we thank you for your insight and time spent improving our work.

Reviewer 2 Report

The paper presents a model for the prediction of soil respiration from plant productivity taken from remote sensing sensors in the sub-arctic region. In my opinion, the paper develops an original contribution to the field of regional carbon budget with several data to support the findings. So I recommend publishing the article in the present form with no further changes. 

Author Response

The paper presents a model for the prediction of soil respiration from plant productivity taken from remote sensing sensors in the sub-arctic region. In my opinion, the paper develops an original contribution to the field of regional carbon budget with several data to support the findings. So I recommend publishing the article in the present form with no further changes. 

Thank you for taking the time to offer your expert review.

Reviewer 3 Report

Abstract: Too much emphasis is placed on the introduction in the summary, and materials and methods are lacking. There is also too little reporting of the results. Keywords: Do not repeat words from the title Introduction: In the introduction itself, it would be important to provide an overview of the research on the application of Predicting Soil Respiration from Plant Productivity (NDVI), if any exists. Materials and Methods line 164 – missing reference for method line 179 - I Would ask the authors to make/provide a reference “to fine roots as fixed effects on soil respiration”

Author Response

Abstract: Too much emphasis is placed on the introduction in the summary, and materials and methods are lacking. There is also too little reporting of the results.

After a complete rewrite of the abstract, we believe that it now shows a good mix of background to the research, brief description of the methodology, and key findings. It now reads (Line 8-20): “Soils represent the largest store of carbon in the biosphere with soils at high latitudes containing twice as much carbon (C) than the atmosphere. High latitude tundra vegetation communities show increases in the relative abundance and cover of deciduous shrubs which may influence net ecosystem exchange of CO2 from this C-rich ecosystem. Monitoring soil respiration (Rs) as a crucial component of the ecosystem carbon balance at regional scales is difficult given the remoteness of these ecosystems and the intensiveness of measurements that is required. Here we use direct measurements of Rs from contrasting tundra plant communities combined with direct measurements of aboveground plant productivity via Normalised Difference Vegetation Index (NDVI) to predict soil respiration across four key vegetation communities in a tundra ecosystem. Soil respiration exhibited a nonlinear relationship with NDVI (y = 0.202 e 3.508 x, p< 0.001). Our results further suggest that NDVI and soil temperature can help predict Rs if vegetation type is taken into consideration. We observed, however, that NDVI is not a relevant explanatory variable in the estimation of SOC in a single-study analysis.”

Keywords: Do not repeat words from the title

This was indeed an oversight. We now amended the keyword list. It now reads (Line 23): “Abisko; CO2 flux; LAI; modelling; plant functional type; organic carbon; SOC; vegetation index”.

Introduction: In the introduction itself, it would be important to provide an overview of the research on the application of Predicting Soil Respiration from Plant Productivity (NDVI), if any exists.

Our introduction outlines a number of knowledge gaps. We clearly highlighted the unbalanced body of research between below and aboveground productivity (e.g. see Lines 62-64: “Conversely, the focus of research has been mostly on the visible effect of a warming climate or how this reflects on aboveground productivity [16,33,34]. Our study forms an original contribution to the field of regional carbon budget by attempting to link NDVI to belowground processes, which is a field considerably undeveloped at present.

Materials and Methods: line 164 – missing reference for method

We have added a classic reference for the loss-on-ignition method (Ball, D.F., 1964. Loss‐on‐ignition as an estimate of organic matter and organic carbon in non‐calcareous soils. Journal of soil science, 15(1), pp.84-92.). (Now in line 168).

Line 179 - I Would ask the authors to make/provide a reference “to fine roots as fixed effects on soil respiration”

We have moved reference to software and fixed effects modelling to the end of the sentence to avoid confusion. The section now reads: “… to understand the effect of NDVI, soil moisture, soil temperature, fine roots as fixed effects on soil respiration [49,50].” (Lines 186/187).

We thank you for your comments to help improve our work.

Reviewer 4 Report

Comments on Predicting soil respiration from plant productivity (NDVI) in a sub-arctic tundra ecosystem:

  • Add some of the most important quantitative results to the Abstract.
  • In the last paragraph of the Introduction, the authors should clearly mention the weakness point of former works (identification of the gaps) and describe the novelties of the current investigation to justify us the paper deserves to be published in this journal.
  • Lines 24-27 cite also this relevant paper:

Global surface temperature: A new insight

  • Cite this recent useful paper on the importance of NDVI to improve the literature and to show the importance of your work:

NDVI and Fluorescence Indicators of Seasonal and Structural Changes in a Tropical Forest Succession

  • Table 1 looks like a figure. Please improve the quality.
  • Discuss more on the soil organic carbon and vegetation indices constraints.

Author Response

Add some of the most important quantitative results to the Abstract.

Thank you for your comments. We added information about the non-linearity of Rs/NDVI, which now reads (Lines 20/21): “Soil respiration exhibited a nonlinear relationship with NDVI (y = 0.202 e 3.508 x, p< 0.001).”

In the last paragraph of the Introduction, the authors should clearly mention the weakness point of former works (identification of the gaps) and describe the novelties of the current investigation to justify us the paper deserves to be published in this journal.

Our introduction outlines a number of knowledge gaps based on a review of relevant publications (e.g. lines 59-62; 76-79). These are not always based on weaknesses of former works, but we believe still provide a robust rationale to advance research into the relationship between NDVI and belowground C dynamics.

Lines 24-27 cite also this relevant paper: Global surface temperature: A new insight

We thank you for the suggested references - we used the Valipour et al 2021 paper in the Introduction (line 31): “…growing three times faster than the global mean surface temperature [9].”.

Cite this recent useful paper on the importance of NDVI to improve the literature and to show the importance of your work: NDVI and Fluorescence Indicators of Seasonal and Structural Changes in a Tropical Forest Succession

The suggested paper compares NDVI to a different vegetation index, relevant to atmospheric conditions over tropical forests (with high aerosol densities) and related to chlorophyll fluorescence. Whilst this is indeed an interesting method, it has no direct relevance to our objectives, and we would prefer to keep the text focussed on the relationship between NDVI and belowground C dynamics.

Table 1 looks like a figure. Please improve the quality.

 Thanks for bringing this to our attention. We included a new table (Line 210/211).

Discuss more on the soil organic carbon and vegetation indices constraints.

We wanted to keep this discussion focused strictly on the ecosystem under observation and within the plant health index that we selected for the study and for which we have direct evidence. We would welcome, however, any insight you would see fit to improve this section.

Round 2

Reviewer 1 Report

The authors have addressed the majority of previous concerns. Minor concerns and missed details of descriptions have been added.

Author Response

Dear reviewer,

Thank you for your time and expertise.

Kind regards,

Olivia

Reviewer 3 Report

The authors have respects all the comments i gave as reviewer and corrected everything needed. 

Author Response

(The authors gave the same response as above.)

Reviewer 4 Report

I appreciate the authors addressing the comments. The quality of the manuscript is acceptable now. Congrats!

Author Response

(The authors gave the same response as above.)
